# Cascaded metasurfaces for high-purity vortex generation

Feng Mei [1,7], Geyang Qu [2,7], Xinbo Sha [1], Jing Han[1], Moxin Yu[1], Hao Li[1], Qinmiao Chen [1], Ziheng Ji [1], Jincheng Ni[3], Cheng-Wei Qiu[3], Qinghai Song [1,2] ✉, Yuri Kivshar [4,5] ✉ & Shumin Xiao [1,2,6] ✉

We introduce a new paradigm for generating high-purity vortex beams with metasurfaces. By applying optical neural networks to a system of cascaded phase-only metasurfaces, we demonstrate the efficient generation of high-quality Laguerre-Gaussian (LG) vortex modes. Our approach is based on two metasurfaces where one metasurface redistributes the intensity profile of light in accord with Rayleigh-Sommerfeld diffraction rules, and then the second metasurface matches the required phases for the vortex beams. Consequently, we generate high-purity $LG_{p,l}$ optical modes with record-high Laguerre polynomial orders $p = 10$ and $l = 200$, and with the purity in $p$, $l$ and relative conversion efficiency as 96.71%, 85.47%, and 70.48%, respectively. Our engineered cascaded metasurfaces suppress greatly the backward reflection with a ratio exceeding −17 dB. Such higher-order optical vortices with multiple orthogonal states can revolutionize next-generation optical information processing.

The physics of optical vortex beams carrying orbital angular momentum (OAM) has been explored intensively due to its promising applications in high-capacity optical communications and quantum optics[1–5]. A series of techniques have been developed for generating optical vortices with spiral phase plates, holograms, spatial light modulators, modulated microcavities,[6–9] and more recently with optical metasurfaces[10–13]. The flat profile of the metasurface is essential to minimize and integrate the OAM beam generator. Its subwavelength unit size can greatly increase the resolution, crucial for high-quality vortices. Recently, vortex beam with topological charge up to $l = 100$ and multiplexed vortex beam array have been successfully produced with metasurface J-plate and vortex metalenses, respectively[14–22]. Despite this progress, typical metasurfaces only consider the phase modulations. Under the illumination of Gaussian beams, the modulated outputs are not the eigenmodes in quadratic index media, and they leak to other radial orders, resulting in relatively low purity, and thus restricting their practical applications in communications,

quantum random walks, as well as the high-precision interferometric experiments[23,24].

In the past decade, several approaches have been demonstrated to improve the purity of vortex beams. For instance, the small aperture and mode cleaner cavity can effectively filter out the unwanted radial modes[24]. The implementation of metasurface phase plate or nanoparticles into a laser cavity has also shown the potential of producing high-purity optical vortices[14,25,26]. These techniques, however, require relatively bulky systems, opposite to the recent trends in miniaturization and integration. Very recently, the simultaneous control of amplitude and phase with metasurfaces has triggered new opportunities to realize high-purity OAM beams[27,28]. Nonetheless, the overall system is still relatively bulky and the modulation on amplitude induces a perceptible loss in beam power[28]. Up to now, the efficient realization of high-purity vortex beams in a compact form is still missing. Recently, the emergence of cascaded metasurfaces has expanded the freedom of the optical field, making multifunctional

[1]Ministry of Industry and Information Technology Key Lab of Micro-Nano Optoelectronic Information System, Harbin Institute of Technology Shenzhen, 518055 Shenzhen, P. R. China. [2]Pengcheng Laboratory, 518055 Shenzhen, Guangdong, P. R. China. [3]Department of Electrical and Computer Engineering, National University of Singapore, 117583 Singapore, Singapore. [4]Nonlinear Physics Center, Research School of Physics, Australian National University, Canberra ACT2601, Australia. [5]Qingdao Innovation and Development Center, Harbin Engineering University, 266000 Qingdao, Shandong, P. R. China. [6]Collaborative Innovation Center of Extreme Optics, Shanxi University, 030006 Taiyuan, Shanxi, P.R. China. [7]These authors contributed equally: Feng Mei, Geyang Qu. ✉e-mail: qinghai.song@hit.edu.cn; yuri.kivshar@anu.edu.au; shumin.xiao@hit.edu.cn

purity metasurfaces possible[29–32]. Herein we utilize the cascaded metasurfaces to produce ultra-high-purity vortex modes without significant power loss.

## Results

We consider a Gaussian beam propagating through two metasurfaces for vortex generation, as shown in Fig. 1a. In a general form, the vortex is described by the Laguerre-Gaussian (LG) mode whose field can be presented as

$$LG_{p,l} = \frac{1}{w_z}\left(\frac{r\sqrt{2}}{w_z}\right)^{|l|} L_p^{|l|}\left(\frac{2r^2}{w_z^2}\right) e^{-\left(\frac{r}{w_z}\right)^2} e^{-il\varnothing} \qquad (1)$$

Here $w_z = w_0\sqrt{1+\left(\frac{z\lambda}{\pi w_0^2}\right)^2}$ and $w_O$ is the beam waist and $L_p^{|l|}$ is the Laguerre polynomial of orders $p$ ($p \geq 0$) and $l$, which are the indices along radial direction and azimuthal direction[23]. The required phase and amplitude distributions of $LG_{p,l}$ vortex modes[28] are realized with two cascaded metasurfaces engineered with the help of the inverse design[33,34]. The field distribution of one $LG_{p,l}$ vortex is defined as the target $\vec{E}(\vec{x}_{target})$, which is converted to the phase profiles in metasurface-1 ($\vec{S}_{meta-1}(\vec{x}_1)$) and metasurface-2 ($\vec{S}_{meta-2}(\vec{x}_2)$) following the equation

$$\vec{E}\left(\vec{x}_{target}\right) = \int_\Sigma \left( \int_\Sigma \vec{E}_{source}(\vec{x}_1) \exp\left(j \cdot \vec{S}_{meta-1}(\vec{x}_1)\right) H(\vec{x}_1,\vec{x}_2) dx_1 \right) \\ \exp\left(j \cdot \vec{S}_{meta-2}(\vec{x}_2)\right) H(\vec{x}_2,\vec{x}_{target}) dx_2 \qquad (2)$$

where $\vec{E}_{source}$ is the source field and $H$ is the Rayleigh-Sommerfeld diffraction theory describing the field propagation between two locations ($\vec{x}$). The entire design flow is also illustrated in Fig. 1a. A stochastic gradient descent algorithm, Adam, has been employed to back-propagate the errors and update the metasurfaces to minimize the loss function[35]. Starting with a random meta-design, as depicted in Fig. 1b, the iterations of the optimization loops reduce the mean square error (MSE) and the device performances converge to the design criteria.

To fully illustrate the advantages of our strategy, we take the $LG_{10,200}$ mode as an example and design the metasurfaces. The phase profiles of two metasurfaces have been optimized and shown in Fig. 2a, b (see details in Supplementary Note 1). Under the illumination of a Gaussian beam, the cascaded metasurfaces show the capability of converting incident light to a donut-shaped beam. Different from the strong spreading of beams with conventional approaches, the profiles of beams generated by the cascaded metasurfaces are well preserved with increasing distance behind the second layer. The purity of the output is analyzed by exploiting the orthonormality of LG modes. The corresponding weighting coefficient has been calculated with $c_{p,l} = \langle U(r)|LG_{p,l}(r)\rangle$, where $U(r)$ represents an arbitrary LG vortex mode[15]. Figure 2c shows the purity of vortex of Azimuthal index. Almost 100% of the transmitted power is distributed at $l = 200$. The weighting coefficients on other Azimuthal orders are all suppressed to -0. Similarly, the purity along the radial direction has also been examined and plotted in Fig. 2d. The power distributing at $p = 10$ is more than 99.72%. The power distributions on the other radial orders are negligibly small. All these results clearly confirm the capacity of the cascaded metasurfaces to produce the high-purity LG vortex.

The intensity distributions and the corresponding phase profiles at different axial positions have been simulated and summarized in Fig. 2e–h. When light passes metasurface-1, its intensity profile is the same as the incident one (see Supplementary Note 1), whereas the phase distribution is strongly modulated (Fig. 2e). With the propagation of the optical field to the second metasurface, the intensity pattern becomes 11 concentric rings. Meanwhile, numerous noises rise in the phase profile in Fig. 2f and deviate the phase shift from the designed value for a perfect $LG_{10,200}$ mode. Once the light transmits the second metasurface, the intensity distribution is kept as concentric rings, but the overall phase front has been compensated back to the designed values (see Fig. 2g). The control of intensity distribution and phase profile produces the designed LG vortex mode, which can be well preserved at far field in Fig. 2h.

Then the underlying mechanism of the cascaded metasurfaces becomes very clear. The first metasurface mostly changes the intensity profile to a donut shape, and the second one compensates the phase shifts to realize the designed helical phase front. In our system, two

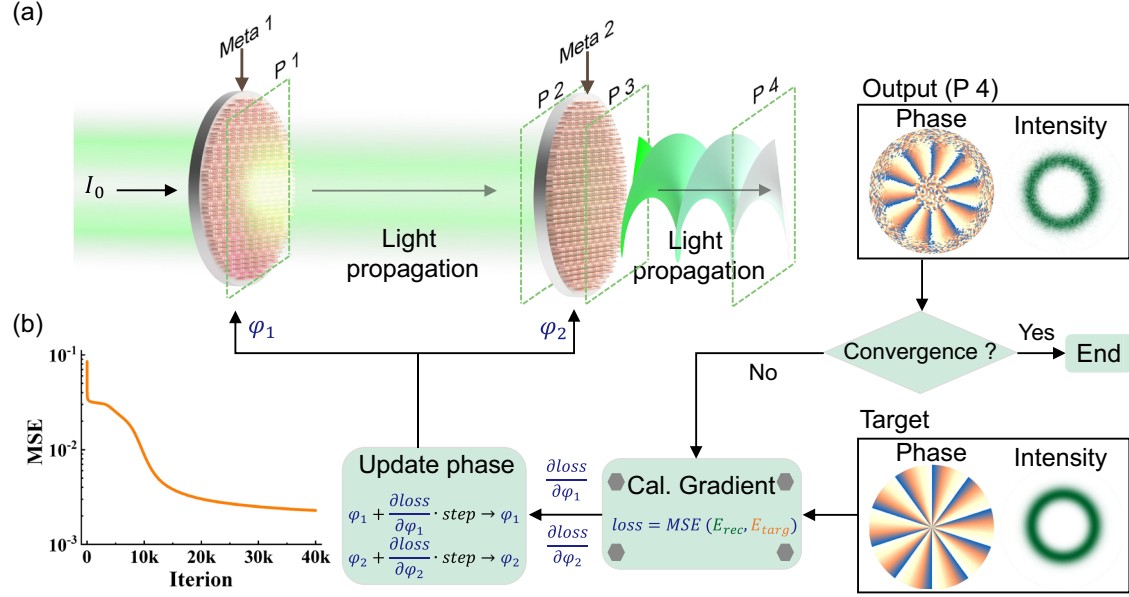

**Fig. 1 | Working principle and inverse design realized with cascaded metasurfaces. a** The schematic of the cascaded metasurfaces composed of numerous TiO$_2$ nanorods. With the propagation of light, the intensity distribution and the phase profile are modulated by two phase-only metasurfaces in sequence. The flow chart of the design for large-scale cascaded metasurfaces is also implemented. **b** Started with a random phase, the optimization loop eventually converses to the target fields with a very small MSE.

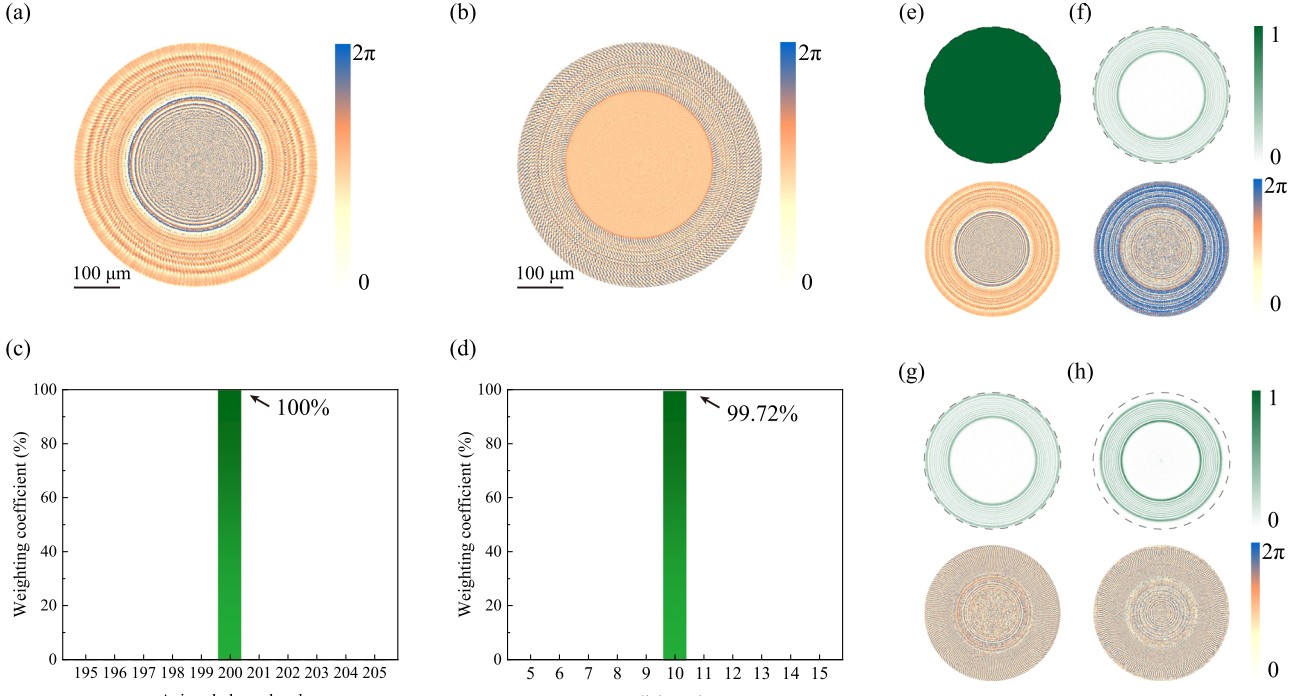

**Fig. 2 | Numerically designed metasurfaces for the generation of the $LG_{10,200}$ mode. a** and **b** are the numerically calculated phase profiles for metasurface-1 and metasurface-2. **c** and **d** show the purity of vortex beam in Azimuthal index and radial index. **e–h** corresponds to the intensity profiles (top) and the phase distributions of modulated light at position P1-P4 marked in Fig. 1a. In our design, two metasurfaces are placed face-to-back, with an intermediate area containing 1.1 mm thick silica substrate and 0.2 mm air gap.

layers are both phase-only metasurfaces. The donut-shaped intensity profile is realized with the field re-distribution following the Rayleigh-Sommerfeld diffraction theory instead of direct modulation on amplitude at each pixel[36]. As a result, most of the incident power should be preserved by the cascaded metasurfaces, and a high conversion efficiency can be expected. Our numerical calculation shows that 88.31% power of incident light has been converted to the vortex beam. This is intrinsically different from the simultaneous modulation of amplitude and phase, which typically sacrifices the incident power perceptibly (see Supplementary Note 2). This physical model demonstrates that the cascaded metasurfaces can achieve complex field modulation with high efficiency. The same mechanism can be extended to a series of nanophotonic devices using several algorithm optimization methods.

Based on the above analysis and numerical simulation, we have fabricated the designed two metasurfaces in 1000 nm $TiO_2$ films with a combined process of electron-beam lithography and reaction ion etching (see "Methods" and Supplementary Note 3)[37]. Figure 3a, b are top-view microscope images of two metasurfaces. They have circular shapes with diameters of 532 μm, and the internal patterns follow the phase profiles in Fig. 2 well. The insets depict high-resolution tilt-view scanning electron microscope (SEM) images of two metasurfaces. They are composed of $TiO_2$ nanorods with square cross-sections and vertical sidewalls. The sizes of the nanorods have been numerically optimized to match the desired phases of metasurfaces while maintaining high transmittance (see "Methods" and Supplementary Note 3).

The two metasurfaces are then aligned in an optical setup and characterized under the illumination of a 532 nm continuous-wave (CW) laser diode (see "Methods" and Supplementary Note 4). The transmitted beam profile behind the cascaded metasurfaces is recorded with a CCD camera. The top inset in Fig. 3d shows the recorded beam profile projected to infinity by an objective lens. It is composed of concentric rings and negligible diffusion to the background. The

enlarged microscope image (bottom inset in Fig. 3d) reveals that the donut is composed of 11 rings, consistent with the designed radial index $p = 10$ well.

To accurately determine the purity of the vortex, a spatial light modulator has been applied to determine the weighting coefficients for different azimuthal and radial orders[15]. The experimental results are summarized in Fig. 3c, d. For the Azimuthal index, approximately 85.47% of the incident power is converted to the state of $l = 200$. The rest is mostly distributed at $l = 198$, 201 and 202 with a ratio of 5.81%, 2.91%, and 5.23%, respectively. In the radial direction, we find that 96.71% of the transmitted power has been converted to the state of $p = 10$. The spreading to $p = 9$ state is about 3.29% and the rests are negligibly small. The conversion efficiency, the ratio of the power of the donut beam over the transmitted power, has also been characterized. The experimentally recorded efficiency is about 70.48%. All these experimental observations are consistent with the numerical calculations (orange dots) and clearly demonstrate the capability of our cascaded metasurfaces to convert incident Gaussian beam to a well-defined LG vortex mode. Note that radial order $p = 10$ and Azimuthal order $l = 200$ are the record values for integrated vortex generation.

Interestingly, the above cascaded metasurfaces can go beyond the efficient generation of LG modes. Two metasurfaces are connected by the Rayleigh-Sommerfeld diffraction theory, which is optimized for forward propagation only. For the reflected backward waves, it shall pose a huge loss and thus efficiently suppress their propagation. Practically, the cascaded metasurfaces are utilized as external cavity devices. As a result, this characteristic is essential to stabilize the laser system. We have tested this possibility by placing a mirror behind the second metasurface (inset in Fig. 3e). Figure 3e shows that the power of reflected $LG_{10,-200}$ mode reduces exponentially after passing through two metasurfaces. The recorded intensity profiles can only be barely seen at enormous magnification (Fig. 3f). An isolation of 17.01 dB has been experimentally demonstrated. This is consistent with the

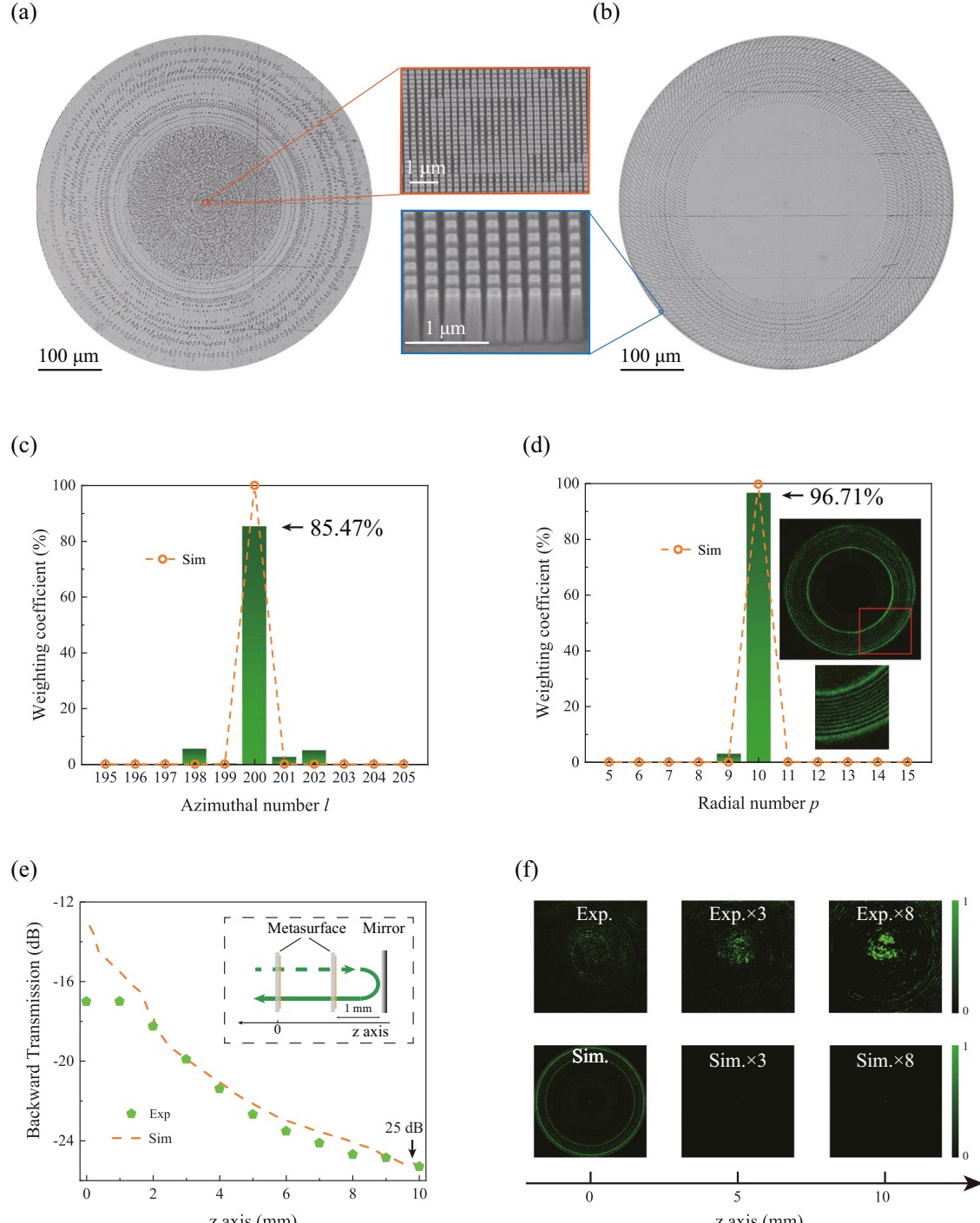

**Fig. 3 | Experimental generation of the $LG_{10,200}$ mode with TiO$_2$ cascaded metasurfaces. a, b** The top-view optical microscope images of two metasurfaces. The insets are their high-resolution tilt-view SEM images. **c** and **d** show the experimentally recorded weighting coefficients along the Azimuthal index (*l*) and radial index (*p*), respectively. The orange dots in (**c**) and (**d**) are the numerically calculated weighting coefficients on Azimuthal and radial indices. The insets in (**d**) display the recorded beam profile at far field. **e** The numerically simulated (dashed line) and experimentally demonstrated (green dots) dependence of backward transmitted power on the propagation distance. **f** The numerically calculated (bottom row) and experimentally recorded (top row) intensity distributions in the x-y plane.

numerical simulation but is not related to the non-reciprocity (see detailed mechanism in Supplementary Note 5).

It is essential to point out that the concept of cascaded metasurfaces is quite generic and applicable to any LG vortex modes. In addition to the $LG_{10,200}$ mode, we have experimentally repeated the generation of other LG modes, e.g., $LG_{0,10}$ and $LG_{3,100}$ modes (see detailed results in Supplementary Notes 6 and 7). As depicted in Table 1, we show that all these LG modes can be produced with high purity and high efficiency. The overall purity $P_{Azimuthal} \cdot P_{Radial}$ are very close in all these experiments, which is lower than the numerical simulation. The slight reductions in purity and efficiency in the experiment are mainly attributed to the misalignment between two metasurfaces and the fabrication deviations. Among these factors, lateral displacement and rotation between two metasurfaces are critical for high efficiency

**Table 1 | Purity and efficiency of LG modes produced by cascaded metasurfaces**

| LG mode | Diameter | Purity in $l$ | Purity in $p$ | Overall purity | Efficiency |
|---------|----------|---------------|---------------|----------------|------------|
| $LG_{0,10}$ | 266 µm | 98.15% | 87.44% | 85.82% | 82.00% |
| $LG_{3,100}$ | 532 µm | 90.81% | 92.31% | 83.83% | 78.04% |
| $LG_{10,200}$ | 532 µm | 85.47% | 96.71% | 82.66% | 70.48% |

and purity, while gap distance variation also appears to be relatively robust. The influence of random scattering induced by nanoparticles during the fabrication process is limited. A more detailed discussion is given in the Supplementary Note 8. Notably, implementing a global design process that considers all possible experimental errors helps to address the challenge in alignment. By directly integrating two meta-surfaces into a single chip, all the alignment problems can be overcome by the mature nano-alignment technology. As depicted in Supplementary Note 9, the monolithically integrated cascaded meta-surface shows higher compactness and higher conversion efficiency.

In summary, we have proposed and demonstrated experimentally the cascaded metasurfaces for the generation of high-purity vortex beams. With such a compact system, both the purity of vortex and conversion efficiency can be optimized simultaneously. The generated $LG_{10,200}$ vortex modes show purity in the radial direction and conversion efficiency of 96.71% (99.72% in simulation) and 70.48% (88.31% in simulation), far superior to the state-of-the-art of a single metasurface being also comparable to bulky intracavity systems. The ability of high-purity vortex generation, combined with its compact footprint and suppression of backward reflection, makes such cascaded meta-surfaces particularly interesting for integrated optical systems[38]. The concept of cascaded metasurfaces can be extended to multiple cascaded metasurfaces, and potentially it may expand substantially the range of metadevice applications.

## Methods

### Numerical simulation
For the inverse design, the training and testing processes are finished by Python v3.8.0 and Tensorflow v2.4.0 (Google Inc.) The optical responses of each nanopillar, such as transmittance and phase shift, are numerically calculated with a commercial software (COMSOL Multiphysics). The in-plane periodic boundary condition and vertical perfect matching layers are applied. The first one is used to mimic the periodic nanostructures and the latter ones absorb the out-going waves.

### Device fabrication
The metasurfaces are fabricated in 1000 nm $TiO_2$ with a combined process of electron-beam lithography and reactive ion etching. The $TiO_2$ film is coated by electron-beam evaporation (Syekey) onto a 13 nm ITO-coated glass substrate. The deposition rate is 0.065 nm/s. The inversed nanostructures are patterned with an electron beam aligner (Raith E-line) within 200 nm electron beam resist (PMMA A2) and transferred to Cr patterns via the lift-off process. Then the samples are etched with $SF_6$ and $Cl_2$ in a reactive ion etching process (Oxford). The metasurfaces are achieved after removing the Cr mask with Cr etchant.

### Optical characterization
The setups for optical characterizations are shown in Supplementary Note 4. Three setups have been utilized to characterize the intensity profile, interference pattern, and purity of beams. In experiment, the optical alignment is realized by adjusting two metasurfaces to fully match all the markers on two substrates. The alignment accuracy is below 1 micron and can preserve the performances of cascaded metasurface well.

**Reporting summary**
Further information on research design is available in the Nature Portfolio Reporting Summary linked to this article.

## Data availability
The data that support the findings of this study are available on request from the corresponding authors.

## Code availability
The code that supports the findings of this study is available on request from the corresponding authors.

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

## Acknowledgements

This research was supported by National Key Research and Development Program of China (Grant Nos. No. 2022YFA1404700, 2021YFA1400802), National Natural Science Foundation of China (Grant Nos. 12334016, 6233000076, 12025402, 62125501, 12261131500, 92250302, 11934012), New Cornerstone Science Foundation through XPLORER PRIZE, Shenzhen Fundamental Research Project (Grant Nos. JCYJ20210324120402006, JCYJ20220818102218040, JCYJ20200109112805990, GXWD20220817145518001), and Fundamental Research Funds for the Central Universities (Grant Nos. 2022FRRK030004). C.-W.Q. acknowledges the support of the AME Individual Research Grant (IRG) funded by A*STAR, Singapore (Grant No. M22K2c0088, with NUS WBS No. A-8001322-00-00). C.-W.Q. acknowledges financial support from the NRF, Prime Minister's Office, Singapore, under the Competitive Research Program Award (NRF-CRP22-2019-0006).

## Author contributions

Q.S., S.X., and Y.K. conceived the idea and supervised the research. F.M. and G.Q did the design. F.M., X.S., J.H., M.Y., H.L., and Q.C. fabricated the samples. F.M., G.Q. and Z.J. performed the experimental measurements. S.X., Y.K., Q.S., C.-W.Q. and J.N. analyzed the results. All the authors discussed the contents and prepared the manuscript.

## Competing interests

The authors declare no competing interests.
