## [Peer Review File · Nature Communications]

REVIEWER COMMENTS

Reviewer #1 (Remarks to the Author):

The authors of this manuscript propose and demonstrate a new method for generating high-purity vortex beams using cascaded metasurfaces. In this work, the authors utilize two cascaded metasurfaces, each of which is inverse-designed simultaneously to exhibit minimal mean squared error (MSE). The discussion is well-written and organized, with a clear and concise explanation of the proposed method and experimental results. The authors provide a detailed description of the design and fabrication of the cascaded metasurfaces, as well as a thorough analysis of the generated vortex beams' purity and efficiency.

One significant strength of this work is that the authors provide a thorough explanation of the physical mechanisms underlying the proposed method, making it accessible to a broad audience of researchers in the field. The proposed method relies on the phase modulation of incident light by a cascaded system of metasurfaces. This results in the generation of high-purity vortex beams with excellent efficiency. They find that the generated vortex beams exhibit high purity, with negligible contributions from other modes. The authors also measure the conversion efficiency of the metasurface and find it to be high, with minimal losses due to fabrication errors. The authors further demonstrate that the cascaded metasurfaces can suppress backward reflection, making them particularly interesting for cascaded optical systems.

However, the authors should mention the limitations of the experimental arrangements that may impact the purity and efficiency of the generated vortex beams. For example, misalignment between the two metasurfaces and fabrication errors could potentially reduce the overall purity and efficiency of the generated vortex beams. Therefore, it is essential to be mindful of these potential degradation possibilities when designing and implementing this method in future applications.

Overall, I believe that this manuscript makes a significant contribution to the field of beam optics and should be considered for publication in Nature Communications.

Reviewer #2 (Remarks to the Author):

In this paper, the authors proposed a design methodology for high-purity vortex beams based on cascaded metasurfaces. By simultaneously optimizing the phase distributions on the two layers, high-purity LG optical modes with Laguerre polynomial orders $p=10$ and $l=200$ is numerically and experimentally demonstrated. Besides, the authors also carried out the generation of other LG modes that all exhibit high purity. I found the paper and the idea behind it interesting and the structure of the paper is acceptable overall. The simulated and measured results also make sense. However, there are some serious aspects that need improvements and some ambiguities that need to be rectified.

1. The authors stated that the efficient realization of high-purity vortex beams in a compact form is still missing and the proposed design can solve this problem. However, although the efficiency of the proposed devices is larger than previous works, the authors do not illustrate the compactness of their design. In fact, I do not see the required distance between adjacent metasurfaces in the current work, which is quite important. The authors mentioned that the same performance can also be implemented by single-layered metasurface with 1% efficiency, they should also compare the compactness between the single and cascaded design. For many applications involving metasurfaces, compactness may be much important than working efficiency.

2. The authors termed their design process as inverse design and cited a paper [31] discussing deep learning. However, the proposed design method is similar to GS algorithm that is quite different from deep learning. Besides, it seems that the design method in ref [30] is similar to the proposed work, the authors should discuss their difference.

3. The authors mentioned several times that their design process can achieve record-high Laguerre polynomial orders to show the advantages of their work. I am wondering why previous works cannot achieve this goal or they just do not choose such orders? Besides, whether such "record" is valuable for the researchers in this field?

4. The influences of alignment on performances should be further discussed in the main text instead of in the Supporting Information to help the readers to understand the weakness of this method.

5. The authors should further explain why the samples for different Laguerre polynomial orders are designed with different diameters and some theoretical analysis should be given.

6. Cascaded metasurfaces for enhanced or multifunctional electromagnetic manipulation have been widely reported and the following references may be helpful. [10.1038/s41377-019-0193-3](https://doi.org/10.1038/s41377-019-0193-3), [10.29026/oea.2023.220073](https://doi.org/10.29026/oea.2023.220073), [10.1126/sciadv.abf9718](https://doi.org/10.1126/sciadv.abf9718).

Reviewer #1:

We thank the reviewer for the careful review and valuable suggestion. Based on his/her suggestions and comments, we have carefully modified our manuscript. The details can be seen below.

Comment 1: However, the authors should mention the limitations of the experimental arrangements that may impact the purity and efficiency of the generated vortex beams. For example, misalignment between the two metasurfaces and fabrication errors could potentially reduce the overall purity and efficiency of the generated vortex beams. Therefore, it is essential to be mindful of these potential degradation possibilities when designing and implementing this method in future applications.

Our response: We appreciate the valuable comments from the reviewer. As mentioned in the main text (Line-202, Page-7), the experiment performance is deteriorated due to the misalignment between two metasurfaces and imperfect fabrication. Supplementary Note-8 also discusses the effect of sample rotation. However, as the reviewer pointed out, a more comprehensive discussion on the limitations of the experimental arrangements should be included in order to further promote our method. Fortunately, all these misalignments can be overcome with optimized numerical designs and modern nanofabrication technologies. Therefore, high performance meta-devices can still be achieved experimentally.

The misalignment between two metasurfaces comes from the lateral displacement, angle variation, gap distance, and the tilted angle. Generally, lateral displacement between these two metasurfaces is the most common type. To quantify the effect of the lateral displacement, the normalized displacement δ_x is defined as $\delta_x = \Delta x/D$, where Δx is the lateral displacement and D is the diameter of the metasurface. By taking the design in the main text as an example ($l = 200$, $p = 10$), the second metasurface is shifted along the x-axis and the 45° direction (insert in Fig. R1(a)) while keeping other parameters unchanged. Figure R1(a) shows the intensity and the phase distribution under different displacement number δ_x along x-axis. In order to have a clearer view of the optical field and phase distribution, we have extracted partial area (with a size of $78 * 78 \mu\text{m}$) of the annular field distribution on the left side along the centerline of outputs. The following intensity and the phase distribution are all selected from the same region. With the increasing of δ_x , the field distribution of LG mode degrades quickly into random noise. Figure R1(b) and (c) summary the efficiency and the purity variation versus δ_x . The overall purity decreases to 37.36% and 12.66% when δ_x reaches 1.95‰ along the x-axis and 45° direction, respectively. The misalignment number 1.95‰ corresponds to size of $1.04 \mu\text{m}$, which is about 2 pixels of nanopillars. Similar results are observed for the degradation of performance along the y-axis. The above simulations indicate that the preformation of LG modes generation is very sensitive to the lateral displacement.

Figure R1. **Numerical simulation for the effect of lateral misalignment for generation of the $LG_{10,200}$ mode.** (a) shows the intensity profiles (top) and the phase distributions of modulated light. (b) and (c) show the efficiency and purity of vortex beam versus the lateral displacement between two metasurfaces.

The influence of the angle variation by rotating the second metasurface from 0° to 3.6° is also studied. Figure R2(a) shows the intensity and the phase evolution as a function of the rotation angle. Interestingly, a periodic variation can be observed with a period of 1.8° . The efficiency and the purity also show similar phenomenon (Fig. R2(b)-(d)). This is attributed to the periodic phase distribution of the metasurface along the azimuthal direction. For the LG mode with topological charge of 200, a rotational symmetry of $360^\circ/200 = 1.8^\circ$ is maintained, resulting in the periodic variation with period of 1.8° . Within a period of variation, the dramatical decrease of the intensity distribution and the purity indicates that the angle variation also plays an important role in LG modes generation.

Figure R2. **Numerical simulation for the effect of angle variation for generation of the $LG_{10,200}$ mode.** (a) shows the intensity profiles (top) and the phase distributions of modulated light. (b), (c) and (d) show the efficiency and purity of vortex beam versus the angle variation between two metasurfaces.

Another misalignment factor arises from the gap distance between these two metasurfaces, as shown in the insert of Fig. R3(a). The normalized displacement δ_{gap} is defined as $\delta_{gap} = \Delta x/d$, where Δx is the displacement and the d is the designed optical distance between the metasurfaces. Figure R3(a) shows the intensity and phase generated by the cascaded metasurfaces with different gap distances. Unlike the previously discussed misalignments, the cascaded metasurfaces system is quite robust to the gap distance. As shown in Fig. R3(c), the purity of LG mode only drops by 14% when δ_{gap} reaches 3.85%. The high robustness comes from the unique mechanism of the cascaded metasurfaces. As mentioned in the main text, the first metasurface contributes to the donut-shaped intensity profile, which varies slowly with the propagating distance.

Figure R3. **Numerical simulation for the effect of gap distance for generation of the $LG_{10,200}$ mode.** (a) shows the intensity profiles (top) and the phase distributions of modulated light. (b) and (c) show the efficiency and purity of vortex beam versus the angle variation between two metasurfaces.

Another misalignment issue comes from the tilted angle. To simulate the tilt variation, the first metasurface is fixed and a linear phase delay φ_θ is added to the second metasurface. The phase delay is defined as $\varphi_\theta(x) = kx\sin(\theta)$, where k is the wave vector. x and θ are the coordinate and the tilted angle of the second metasurface. The added phase delay is kept as constant along y -axis. In the practical experiment, the tilted angle can be controlled precisely within $\pm 0.5^\circ$. Therefore, θ number is kept below 1° in the simulation. Figure R4(a) shows the intensity and the phase profiles generated by the cascaded metasurfaces with different θ . As shown in Fig. R4(b) and (c), the efficiency and the purity maintain high when θ is less than 0.5° . It is not surprising that tilted angle has limited effect on the performance of vortex beams generation. Under small angle conditions, the misalignment between pixels, the change of the propagating distance and phase-response variation of meta-atoms can be almost neglected but only introducing tiny beam deflection, and will not affect the efficiency and the purity of vortex beam.

Figure R4. **Numerical simulation for the effect of titled angle for generation of the $LG_{10,200}$ mode.** (a) shows the intensity profiles (top) and the phase distributions of modulated light. (b) and (c) show the efficiency and purity of vortex beam versus the titled angle between two metasurfaces.

In addition to the misalignment between two metasurfaces, imperfect fabrication will also degrade the preformation of the vortex beam. If the geometries of meta-atoms randomly deviate from the design, phase perturbation will be induced. The random phase perturbation is introduced to both designed phase profile by adding a random matrix from 0 to 1 multiplied by a random phase amplitude δ_φ in the simulation. As shown in Fig. R5, the cascaded metasurfaces are highly robust to random scattering/phase perturbation. The efficiency only drops by 24% when the random phase perturbation even reaches 0.5π . This phase change corresponds to 25 nm dimension change, which is far larger than the experimental bias.

Figure R5. **Numerical simulation for the effect of the imperfect fabrication for generation of the $LG_{10,200}$ mode.** (a) shows the intensity profiles (top) and the phase distributions of modulated light. (b) and (c) show the efficiency and purity of vortex beam versus the angle variation between two metasurfaces.

Overall, the misalignment issue and the imperfect fabrication can be technically addressed by optimizing the fabrication process, such as bonding these two metasurfaces with high alignment accuracy, typically lower than 100 nm. Alternatively, a fabrication/alignment-robust optimization method can be applied in the initial design by incorporating all experimental errors. In the revised manuscript, the robustness analysis is updated in Supplementary Note-8 and more discussion has been added in the main text (Page-7, Line-200) as below in blue.

‘The overall purity $P_{Azimuthal} \cdot P_{Radial}$ are very close in all these experiments, which is lower than the numerical simulation. The slight reductions of purity and efficiency in experiment are mainly attributed to the misalignment between two metasurfaces and the fabrication deviations. Among these factors, lateral displacement and rotation between two metasurfaces are critical for high efficiency and purity, while gap distance variation also appears to be relatively robust. The influence of random scattering induced by nanoparticles during the fabrication process is limited. A more detailed discussion is given in the Supplementary Note-8. However, a global design process of taking all the possible experimental errors into consideration can help to solve the problem. Based on the initial experiment, all the experimental errors have been well controlled by integrating two metasurfaces into a single chip using nano-alignment technology, which successfully verified the feasibility of integrating cascaded metasurfaces with high compactness (see Supplementary Note-9).’

Reply to Reviewer #2:

We thank the reviewer for the careful review and valuable suggestion. Based on his/her suggestions and comments, we have carefully modified our manuscript. The details can be seen below.

Comment 1: The authors stated that the efficient realization of high-purity vortex beams in a compact form is still missing and the proposed design can solve this problem. However, although the efficiency of the proposed devices is larger than previous works, the authors do not illustrate the compactness of their design. In fact, I do not see the required distance between adjacent metasurfaces in the current work, which is quite important. The authors mentioned that the same performance can also be implemented by single-layered metasurface with 1% efficiency, they should also compare the compactness between the single and cascaded design. For many applications involving metasurfaces, compactness may be much important than working efficiency.

Our response: We thank the reviewer for the careful review and the valuable question. In our design, two metasurfaces are placed face-to-back, with an intermediate distance containing a 1.1 mm thick silica substrate and 0.2 mm air gap. Such a relatively big distance is limited by the fabrication issue and the optical path in our design. More detailed parameters are added in Page-4, Line-114 in the main text as below in blue. However, the gap distance can be further decreased and the cascaded metasurfaces can also be integrated together with SU-8 as spacer to ensure the compactness. As shown in Fig. R6(a), a series of integrated cascaded metasurfaces using SU-8 as the spacing layer are designed. The thickness of SU-8 gradually decreases from 1300 μm to 10 μm . Figure R6(b) plots the relations between conversion efficiency and overall purity versus separation distance. Even with the thin spacing layer, the purity of the vortex modes remains close to unity, while the efficiency decreases very slowly to 73.93%. Both of the purity and the efficiency maintain at high levels. To compare the cascaded design with the single complex-modulation metasurface, the conversion efficiency versus compactness (thickness^{-1}) is shown in Fig. R6(c) for both cases. Although the single-layer metasurface has a much more compact form with a thickness of about 1 μm , the conversion efficiency of 1% is too low to meet the requirement of some practical applications. In contrast, the cascaded metasurfaces with 10- μm thickness still maintains a high efficiency of 73.93% and near-unity mode purity.

Figure R6. **Design for the integrated cascaded metasurfaces for generation of the $LG_{10,200}$ mode.** (a) shows the design sketch. (b) shows the efficiency and purity of vortex beam versus the separation distance between two metasurfaces. (c) shows the conversion efficiency versus compactness (thickness⁻¹) for single and cascaded metasurfaces.

The fabrication process for 10- μm -thick integrating cascaded metasurfaces is also initially implemented in the experiment. The brief fabrication process is shown in Fig. R7(a). The first metasurface is fabricated with alignment-markers around the metasurface with 40 μm away, following the same top-down process described in Supplementary Note-3. Then SU-8 photoresist is spin-coated on the first layer under negative pressure to make sure it fully fills the gap of the nanopillars. The thickness of SU-8 layer is controlled by the spin-coating speed. Figure R7(b) shows the optical image of the SU-8 gap layer after soft-bake. The SU-8 spacer has flat surface with smooth roughness with RMS number of 295.7 pm, which is smooth enough to ensure the gap distance and the tilted angle for the metasurfaces. Afterwards, Si_3N_4 film with thickness of 1000 nm is deposited on the SU-8 spacing layer using plasma-enhanced chemical vapor deposition (PECVD). To fabricate the second metasurface, the key point is to align it to the first metasurface. The alignment process is conducted under high-resolution microscope using the following method. First, metal markers are fabricated through lift-off process on the new deposited Si_3N_4 layer. Since the bottom markers are invisible now for electron beams, the top and bottom markers usually have lateral and rotation misalignments (Figure R7(c)). High-resolution microscope is applied to measure the lateral and rotation misalignments between two markers with resolution up to 250 nm. The alignment shifts are then taken into account and compensated in the GDS layout for the second metasurface in EBL exposure. Figure R7(d) shows the optical images of two metasurfaces after EBL exposure. Two metasurfaces can be perfectly aligned to each other. Further etching of the second metasurface is also performed in our experiment and the SEM image is shown in Fig. R7(e). The nanopillars can maintain the design geometry with very vertical angle. Figure R7(f) shows the final image of the integrated cascaded metasurfaces, which show high compactness.

Figure R7. **Fabrication process for the integrated cascaded metasurfaces.** (a) shows the design sketch. (b) and (c) show the alignment process. (e) shows the SEM image of the second metasurface. (f) shows the final image of the integrated cascaded metasurfaces.

Following the above process, the cascaded metasurfaces prototype can be ready integrated with alignment accuracy around 250 nm, which is limited by the microscope resolution and is precise enough for our experiment. The related contents have been added to the Supplementary Note-9 and in the main text (Page-7, Line-204) as below.

‘The overall purity $P_{Azimuthal} \cdot P_{Radial}$ are very close in all these experiments, which is lower than the numerical simulation. The slight reductions of purity and efficiency in experiment are mainly attributed to the misalignment between two metasurfaces and the fabrication deviations. Among these factors, lateral displacement and rotation between two metasurfaces are critical for high efficiency and purity, while gap distance variation also appears to be relatively robust. The influence of random scattering induced by nanoparticles during the fabrication process is limited. A more detailed discussion is given in the Supplementary Note-8. However, a global design process of taking all the possible experimental errors into consideration can help to solve the problem. Based on the initial experiment, all the experimental errors have been well controlled by integrating two metasurfaces into a single chip using nano-alignment technology, which successfully verified the feasibility of integrating cascaded metasurfaces with high compactness (see Supplementary Note-9).’

The designed detailed parameters of our cascaded metasurfaces have been added to Page-4, Line-114 in the main text as below in blue.

‘In our design, two metasurfaces are placed face-to-back, with an intermediate area containing 1.1 mm thick silica substrate and 0.2 mm air gap.’

Comment 2: The authors termed their design process as inverse design and cited a paper [31] discussing deep learning. However, the proposed design method is similar to GS algorithm that is quite different from deep learning. Besides, it seems that the design method in ref [30] is similar to the proposed work, the authors should discuss their difference.

Our response: We thank the reviewer for this careful review and important question. In our design, the stochastic gradient descent algorithm (SGD) is used, which relies on the error back-propagating and gradient calculation. This method is widely used in inverse design for nanophotonics devices, including diffractive deep neural networks [R1], polarization splitters [R2], and beam deflectors [R3]. By setting an appropriate loss function, such as mean squared error (MSE) between the output field and the theoretical Laguerre-Gaussian (LG) mode, SGD calculates the gradient for both metasurfaces jointly and optimizes the phase profiles to minimize the loss function. However, GS algorithm is only used for phase retrieval problems using local intensity compensation and does not involve the concept of gradients. Therefore, our inverse design method is fundamentally different from the GS algorithm. Furthermore, the design objective of this work is to control the intensity and the phase at the same time, while GS algorithm only controls the intensity. Generation of a high purity LG mode with high efficiency requires to control the intensity and the phase simultaneously. As a result, SGD-like algorithms should be used in this work.

While the Ref. [30] (Ref. [34] in the revised manuscript, Opt. Express 27, 30308-30331 (2019)) shares a similar design method with this work, it should be noted that this work focuses on physical mechanism, numerically and experimentally realization of high purity vortex beam. Especially, the underlying physical mechanism is obtained and the design can go beyond algorithm optimization. As mentioned in the main text, the first metasurface contributes to the intensity modulation for the donut shape, while the second metasurface contributes to the phase compensation. This clear physical model demonstrates that the cascaded metasurfaces can achieve complex field modulation with high efficiency. The same mechanism can be extended to series nanophotonic devices using other algorithm optimization methods.

In the revised manuscript, the focused point of this work is further emphasized in Page-5, Line-137 in main text as below:

‘This physical model demonstrates that the cascaded metasurfaces can achieve complex field modulation with high efficiency. The same mechanism can be extended to series nanophotonic devices using several algorithm optimization methods.’

We also exchange Ref. [30] (Ref. [33] in the revised manuscript) to ‘Molesky, S., Lin, Z., Piggott, A. Y., Jin, W., Vuckovi, J. and Rodriguez, A. W., Inverse design in nanophotonics. Nature Photon. 12, 659-670 (2018)’, which used inverse design method to optimize the phase and amplitude distributions of light at the same time.

Comment 3: The authors mentioned several times that their design process can achieve record-high Laguerre polynomial orders to show the advantages of their work. I am wondering why previous works cannot achieve this goal or they just do not choose such orders? Besides, whether such “record” is valuable for the researchers in this field?

Our response: We express our gratitude to the reviewer for raising this important question. The generation of high order LG modes, especially high topological charge (TC) holds great significance for many areas. The most beneficial applications may be the optical communications. The communication capacity can be dramatically enlarged by utilizing the orthogonal OAM states [R4-R5]. The transmission capacity of quantum communication systems can also be enhanced under high TC [R6]. Besides the information science, high TC beams can also be used to enhance the resolution [R7] and sensitivity [R8] of optical measurements. For optical micro-manipulation, the OAM beam with larger TC can offer a larger operation area due to its bigger radius [R9]. However, all the performance of the applications are highly dependent on the purity of the high order LG modes.

The generation of high-order LG modes is extremely challenging due to the severe energy dispersion in the undesired orders, which resulting in a low purity of desired modes [14, 28]. In order to achieve high purity LG modes, complex modulation of amplitude and phase is typically required. Although several methods have been used, most of them involve bulky systems, and make it unsuitable for optical communications or quantum entanglement experiments. Metasurfaces based on complex modulation of amplitude and phase tackled this problem with ultra-compact form. Nevertheless, most of the works based on metasurfaces up to now only manipulate the phase or the amplitude separately. Either the purity or the efficiency will be limited. In contrast, this work utilizes the cascaded metasurfaces to achieve complex modulation at the same time, resulting in the generation of LG modes with record high Laguerre polynomial orders and high efficiency. Our approach offers obvious advantages in terms of efficiency and compactness, and demonstrate a new method for optical field manipulation in a broader range of applications.

Comment 4: The influences of alignment on performances should be further discussed in the main text instead of in the Supporting Information to help the readers to understand the weakness of this method.

Our response: We thank the reviewer for this valuable question. In the revised main text, further discussions about the influences of the alignment and the imperfect fabrication are added in the main text (Page-7, Line-200). Here is the copy of the related contents.

‘The overall purity $P_{Azimuthal} \cdot P_{Radial}$ are very close in all these experiments, which is lower than the numerical simulation. The slight reductions of purity and efficiency

in experiment are mainly attributed to the misalignment between two metasurfaces and the fabrication deviations. Among these factors, lateral displacement and rotation between two metasurfaces are critical for high efficiency and purity, while gap distance variation also appears to be relatively robust. The influence of random scattering induced by nanoparticles during the fabrication process is limited. A more detailed discussion is given in the Supplementary Note-8. However, a global design process of taking all the possible experimental errors into consideration can help to solve the problem. Based on the initial experiment, all the experimental errors have been well controlled by integrating two metasurfaces into a single chip using nano-alignment technology, which successfully verified the feasibility of integrating cascaded metasurfaces with high compactness (see Supplementary Note-9).’

We also added a more detailed discussion of the experimental errors in Supplementary Note-8 as below.

Supplementary Note-8: The influences of alignment and fabrication on performances

In our experiment, we observed the slight reduction in both of purity of vortex and the conversion efficiency. To understand such kind of reduction, we have simulated the entire system by introducing many potential deviations. We find the alignment between two metasurfaces plays an essential role. The misalignment between two metasurfaces comes from the lateral displacement, angle variation, gap distance, and the tilted angle. Generally, lateral displacement between these two metasurfaces is the most common type. To quantify the effect of the lateral displacement, the normalized displacement δ_x is defined as $\delta_x = \Delta x / D$, where Δx is the lateral displacement and D is the diameter of the metasurface. By taking the design in the main text as an example ($l = 200$, $p = 10$), the second metasurface is shifted along the x -axis and the 45° direction (Fig. S15(a) insert), while keeping other parameters unchanged. Figure S15(a) shows the intensity and the phase distribution under different displacement number δx along x -axis. In order to have a clearer view of the optical field and phase distribution, we have extracted partial area (with a size of $78 * 78 \mu\text{m}$) of the annular field distribution on the left side along the centerline of outputs. The following intensity and the phase distribution are all selected from the same region. With the increasing of δx , the field distribution of LG mode degrades quickly into random noise. Figure S15(b) and (c) summary the efficiency and the purity variation versus δx . The overall purity decreases to 37.36% and 12.66% when δx reaches 1.95‰ along the x -axis and 45° direction, respectively. The number 1.95‰ corresponds to about 2 pixels. Similar results are observed for the degradation of performance along the y -axis. The above simulations indicate that the lateral displacement plays very important role in LG modes generation.

Figure R8 (Figure S15 in the Supplementary Information). Numerical simulation for the effect of lateral misalignment for generation of the $LG_{10,200}$ mode. (a) shows the intensity profiles (top) and the phase distributions of modulated light. (b) and (c) show the efficiency and purity of vortex beam versus the lateral displacement between two metasurfaces.

The influence of the angle variation by rotating the second metasurface from 0° to 3.6° is also studied. Figure S16(a) shows the intensity and the phase evolution as a function of the rotation angle. Interestingly, a periodic variation can be observed with a period of 1.8° . The efficiency and the purity also show similar phenomenon (Figure S16(b)-(d)). This is attributed to the periodic phase distribution of the metasurface along the azimuthal direction. For the LG mode with topological charge of 200, a rotational symmetry of $360^\circ/200 = 1.8^\circ$ is maintained, resulting in the periodic variation with period of 1.8° . Within a period of variation, the dramatic decrease of the intensity distribution and the purity indicates the important role in LG modes generation.

Figure R9 (Figure S16 in the Supplementary Information). Numerical simulation for the effect of angle variation for generation of the $LG_{10,200}$ mode. (a) shows the intensity profiles (top) and the phase distributions of modulated light. (b), (c) and (d) show the efficiency and purity of vortex beam versus the angle variation between two metasurfaces.

Another misalignment factor arises from the gap distance between these two metasurfaces, as shown in the insert of Fig. S17(a). The normalized displacement δ_{gap} is defined as $\delta_{gap} = \Delta x/d$, where Δx is the displacement and the d is the designed optical distance between the metasurfaces. Figure S17(a) shows the intensity and phase generated by the cascaded metasurfaces with different gap distances. Unlike the previously discussed misalignments, the cascaded metasurfaces system is quite robust to the gap distance. As shown in Fig. S17(c), the purity of LG mode only drops by 14% when δ_{gap} reaches 3.85%. The high robustness comes from the unique mechanism of the cascaded metasurfaces. As mentioned in the main text, the first metasurface contributes to the donut-shaped intensity profile, which varies slowly with the propagating distance.

Figure R10 (Figure S17 in the Supplementary Information). Numerical simulation for the effect of gap distance for generation of the $LG_{10,200}$ mode. (a) shows the intensity profiles (top) and the phase distributions of modulated light. (b) and (c) show the efficiency and purity of vortex beam versus the angle variation between two metasurfaces.

Another misalignment issue comes from the tilted angle. To simulate the tilt variation, the first metasurface is fixed and a linear phase delay φ_θ is added to the second metasurface. The phase delay is defined as $\varphi_\theta(x) = kx\sin(\theta)$, where k is the wave vector. x and θ are the coordinate and the tilted angle of the second metasurface. The added phase delay is kept as constant along y -axis. In the practical experiment, the tilted angle can be controlled precisely within $\pm 0.5^\circ$. Therefore, θ number is kept below 1° in the simulation. Figure S18(a) shows the intensity and the phase profiles generated by the cascaded metasurfaces with different θ . As shown in Fig. S18(b) and (c), the efficiency and the purity maintain high when θ is less than 0.5° . It is not surprising that tilted angle has limited effect on the performance of vortex beams generation. Under small angle conditions, the misalignment between pixels, the change of the propagating distance and phase-response variation of meta-atoms can be almost neglected but only introducing tiny beam deflection, and will not affect the efficiency and the purity of vortex beam.

Figure R11 (Figure S18 in the Supplementary Information). Numerical simulation for the effect of titled angle for generation of the $LG_{10,200}$ mode. (a) shows the intensity profiles (top) and the phase distributions of modulated light. (b) and (c) show the efficiency and purity of vortex beam versus the titled angle between two metasurfaces.

In addition to the misalignment between two metasurfaces, imperfect fabrication will also degrade the preformation of the vortex beam. If the geometries of meta-atoms randomly deviate from the design, phase perturbation will be induced. The random phase perturbation is introduced to both designed phase profile by adding a random matrix from 0 to 1 multiplied by a random phase amplitude δ_φ in the simulation. As shown in Fig. S19, the cascaded metasurfaces are highly robust to random scattering/phase perturbation. The efficiency only drops by 24% when the random phase perturbation even reaches 0.5π . This phase change corresponds to 25 nm dimension change, which is far larger than the experimental bias.

Figure R12 (Figure S19 in the Supplementary Information). Numerical simulation for the effect of the imperfect fabrication for generation of the $LG_{10,200}$ mode. (a) shows the intensity profiles (top) and the phase distributions of modulated light. (b) and (c) show the efficiency and purity of vortex beam versus the angle variation between two metasurfaces.

Overall, the misalignment issue and the imperfect fabrication can be technically addressed by optimizing the fabrication process, such as bonding these two metasurfaces with high alignment accuracy, typically lower than 100 nm. Alternatively, a fabrication/alignment-robust optimization method can be applied in the initial design by incorporating all experimental errors.

Comment 5: The authors should further explain why the samples for different Laguerre polynomial orders are designed with different diameters and some theoretical analysis should be given.

Our response: We thank the reviewer for this important question. The sample diameters (D) are determined by the radial and azimuthal number of the LG modes under the premise of ensuring the purity and recognition. The space-bandwidth product of LG modes increases quickly with Laguerre polynomial orders [R10]. To construct high order LG modes, metasurfaces must cover the space-bandwidth product requirement accordingly. While keeping period of meta-atoms unchanged, increasing the sample diameter can enlarge the space-bandwidth product until satisfying the requirement. In other words, the sampling rate in frequency domain equals to $1/D$ [R11], indicating that larger metasurfaces are able to handle precise frequency variations of high order LG modes. Therefore, larger metasurfaces are

necessary for constructing higher order LG modes with same level high purity and efficiency.

To prove this issue in simulation, a series of metasurfaces with different D for $p = 10$, $l = 200$ LG mode are designed. The calculated output fields are shown in Fig. R13(a). With the decreasing of D , severe random noises can be observed in the output field. Eventually, only two adjacent circles can be observed when D is smaller than $66 \mu\text{m}$. The intensity distribution of the output fields along black lines in Fig. R13 (a) is plot in Fig. R13 (b). When D equals to $532 \mu\text{m}$ (adopted in the main text), the intensity profile matches well to the standard LG beam. With the decreasing of D , background noise increases, and contrast between the peaks and dips drops quickly. When D equals to $66 \mu\text{m}$, there is almost no intensity variation along the radial line. The mode purity also decreases for smaller D (Figure R13(c)). In the premise of efficiency and considering the fabrication ability in real experiment, $D = 266/532/532 \mu\text{m}$ are chosen for LG mode with $p = 0/3/10$ and $l = 10/100/200$.

Figure R13. **Simulated analyses for cascaded metasurfaces with different diameters for generation of the $LG_{10,200}$ mode.** (a) shows the intensity profiles for cascaded metasurfaces with different diameters. (b) shows the intensity plot along the black line in (a). (c) shows the overall mode purity under different diameters.

Comment 6: Cascaded metasurfaces for enhanced or multifunctional electromagnetic manipulation have been widely reported and the following references may be helpful. 10.1038/s41377-019-0193-3, 10.29026/oea.2023.220073, 10.1126/sciadv.abf9718.

Our response: We thank the reviewer for the instructive suggestions. The above works of cascaded metasurfaces have been added to the main text in Page-2, Line-59 as Reference 29-31 and more discussion is added as below.

‘Recently, the emergence of cascaded metasurfaces have expanded the freedom of optical field, making multifunctional metasurfaces possible. [29-31]’

References:

- [R1] Lin X., Rivenson Y., Yardimei N. T., Veli M., Luo Y., Jarrahi M., Ozcan A., All-optical machine learning using diffractive deep neural networks. *Science* 361, 1004-1008 (2018).
- [R2] Sell D., Yang J., Doshay S., Yang R., Fan J. A., Large-angle, multifunctional metagratings based on freeform multimode geometries. *Nano Lett.* 17, 3752-3757 (2017).
- [R3] Xu M., Pu M., Sang D., Zheng Y., Li X., Ma X., Guo Y., Zhang R., Luo X., Topology-optimized catenary-like metasurface for wide-angle and high-efficiency deflection: from a discrete to continuous geometric phase. *Optics Express* 29, 10181-10191 (2021).
- [R4] Wang, J., Yang J., Fazal I. M., Ahmed N., Yan Y., Huang H., Ren Y., Yue Y., Dolinar S., Tur M., Willner A. E., Terabit free-space data transmission employing orbital angular momentum multiplexing. *Nat. Photonics* 6, 488–496 (2012).
- [R5] Bozinovic, N., Yue Y., Ren Y., Tur M., Kristensen P., Huang H., Willner A. E., Ramachandran S., Terabit-scale orbital angular momentum mode division multiplexing in fibers. *Science* 340, 1545–1548 (2013).
- [R6] Fickler R., Lapkiewicz R., Plick W. N., Krenn M., Schaeff C., Ramelow S., Zeilinger A., Quantum entanglement of high angular momenta. *Science* 338, 640-643 (2012)
- [R7] Emile O., Emile J. Naked eye picometer resolution in a Michelson interferometer using conjugated twisted beams. *Opt. Lett.* 42, 354-357 (2017).
- [R8] Belmonte A., Torres J. P., *Opt. Lett.* 36, 4437-4439 (2011).
- [R9] Kovalev A. A., Kotlyar V. V., Porfirev A. P. Optical trapping and moving of microparticles by using asymmetrical Laguerre-Gaussian beams. *Opt. Lett.* 41, 2426-2429 (2016).
- [R10] Xiao Y., Tang X., Wan C., Qin Y., Peng H., Hu C., Qin B. Laguerre-Gaussian mode expansion for arbitrary optical fields using a subspace projection method. *Opt. Lett.* 44, 1615-1618 (2019).
- [R11] Goodman J. W. *Introduction to Fourier optics.* Roberts and Company publishers (2005).

REVIEWER COMMENTS

Reviewer #1 (Remarks to the Author):

In their rebuttal letter, the authors tried to address the purity and efficiency degradation issues, which are found (numerically) to be associated with lateral and angular misalignment, alterations in gap distance, and relative tilting. From the data provided in the response, it is apparent that lateral misalignment necessitates an exacting level of precision. With respect to these concerns, I would like to pose several additional inquiries:

The methods in the manuscript notes that the error in misalignment was maintained below 1 μm . Could the authors elucidate the specific steps the authors took to minimize this misalignment error during the actual measurements? Was there a systematic scan of one of the metasurfaces with subsequent measurements of purity and efficiency? It seems that a mere 1 μm misalignment error could induce a reduction in efficiency by approximately 10% and purity by 50%.

The authors detailed in Fig. 2 that "the two metasurfaces are situated back-to-back, with an intermediate area consisting of a 1.1 mm thick silica substrate and a 0.2 mm air gap". However, it seems there was a discrepancy between the conceptual design and the actual implementation of the measurements. Could you elucidate why these differences exist? Additionally, the authors mentioned the potential for integrating the two metasurfaces to mitigate error, yet there is no initial measurement data provided on the integrated metasurfaces (e.g. LG beam profiles, purity or efficiency). Could you clarify this issue?

In the authors' response to the other reviewer, they mentioned the potential benefits of utilizing higher order LG beams for the formation of orthogonal OAM states. However, taking into account the misalignment and fabrication issues discussed above, I harbor doubts concerning the actual orthogonality achieved in practice. I would appreciate further elaboration on this matter.

Reviewer #2 (Remarks to the Author):

The authors have well addressed all my concerns. This work can be accepted for publication in its current form.

Reviewer #1:

In their rebuttal letter, the authors tried to address the purity and efficiency degradation issues, which are found (numerically) to be associated with lateral and angular misalignment, alterations in gap distance, and relative tilting. From the data provided in the response, it is apparent that lateral misalignment necessitates an exacting level of precision. With respect to these concerns, I would like to pose several additional inquiries.

We thank the reviewer for the careful review and valuable suggestion. Based on his/her suggestions and comments, we have carefully modified our manuscript. The details can be seen below.

Comment 1: The methods in the manuscript notes that the error in misalignment was maintained below 1 μm . Could the authors elucidate the specific steps the authors took to minimize this misalignment error during the actual measurements? Was there a systematic scan on CC=e of the metasurfaces with subsequent measurements of purity and efficiency? It seems that a mere 1 μm misalignment error could induce a reduction in efficiency by approximately 10% and purity by 50%.

Our response: We thank the reviewer for this important question. The lateral displacement plays a critical role in the performance. During the experimental measurements, high-precision movement stages and home-made high-resolution microscopy were used to achieve good alignment. During the measurement, the first metasurface is fixed on an angular adjustment stage, and the second metasurface is fixed on a high-precision 6-dimensional movement stage, with freedom of regulation along the x-, y-, z-, rotation, tilted angles along the longitudinal and transverse directions. Two sets of cross-shaped markers were placed on both samples at the corresponding positions at the top-left corner and the bottom-right corner of the metasurfaces.

The tilt angles of both of these samples are adjusted until they are vertical to the incident light. The tilt angle can be precisely controlled within a deviation of $\pm 0.1^\circ$. When the Fresnel reflected light from the substrate are observed, the samples are strictly vertically in the orientation of the incident light. Then, the lateral position and in-plane rotation of the second metasurface were adjusted continuously until the alignment markers match well to these of the first metasurface. A home-made high-resolution microscopy is used to ensure the accuracy. A 20x objective lens with NA of 0.42 (Mitutoyo Plan Apo) is used to collect the images of both samples, resulting in a vertical imaging resolution of 0.7 μm . Since the two markers are separated by a distance of 775 μm and considering the vertical resolution, the rotation imaging resolution is equal to 0.05° . As mentioned above, the second metasurface is placed on a high-precision 6-dimensional movement stage with lateral accuracy of 200 nm (NanoMax, Thorlabs) and in-plane rotation accuracy of 0.04° , which can fully meet the alignment requirements. The lateral position and rotation angle of the second metasurface are adjusted until all markers match perfectly. Through the above alignment process, the lateral displacement deviation, the rotation angle deviation and the tilt angle deviation can be controlled

below $1\ \mu\text{m}$, below 0.1° and below 0.1° , respectively. After this alignment process, slight adjustment will be further performed to make sure the purity and efficiency of the OAM light are already optimal.

Although a real-time characterization system during the alignment process is not used, the fine tuning after the rough adjustment using high-resolution microscopy ensures that we get the best results experimentally. The reproducibility of this alignment process is also very high. We repeated the optical alignment process 8 times, and measured the efficiency and purity for every time. The specific experimental results are shown in **Fig. R1**. The efficiency and the overall purity of these 8 sets maintain at high level, consistent with the results in the main text.

Figure R1. Experimental results for generation $LG_{10,200}$ mode through multiple measurements. (a) shows the efficiency and purity of vortex beam. (The green dashed box corresponds to the data in the main text). (b) shows the LG mode intensity profiles for all number of experiments.

Detailed alignment process is added in the Note-8 of the Supplementary Materials. And we also add the repeating experiments in the Note-8 of the Supplementary Materials.

“For the alignment of the cascaded metasurfaces, high-precision movement stages and home-made high-resolution microscopy were used to achieve good alignment. During the measurement, the first metasurface is fixed on an angular adjustment stage, and the second metasurface is fixed on a high-precision 6-dimensional movement stage, with freedom of regulation along the x-, y-, z-, rotation, tilted angles along the longitudinal and transverse directions. Two sets of cross-shaped markers were placed on both samples at the corresponding positions at the top-left corner and the bottom-right corner of the metasurfaces.

The tilt angle of both of these samples are adjusted until they are vertical to the incident light. The tilt angle can be precisely controlled within a deviation of $\pm 0.1^\circ$. When the Fresnel reflected light from the substrate are observed, the samples are strictly vertically in the orientation of the incident light. Then, the lateral position and in-plane rotation of the second metasurface were adjusted continuously until the alignment markers match well to these of the first metasurface. A home-made high-resolution microscopy is used to ensure the accuracy. A 20x objective lens with NA of 0.42 (Mitutoyo Plan Apo) is used to collect the images of both samples, resulting in a vertical imaging resolution of $0.7\ \mu\text{m}$. Since the two markers are separated by a distance of $775\ \mu\text{m}$ and considering the vertical resolution, the rotation imaging resolution is equal to 0.05° . As

mentioned above, the second metasurface is placed on a high-precision 6-dimensional movement stage with lateral accuracy of 200 nm (NanoMax, Thorlabs) and in-plane rotation accuracy of 0.04° , which can fully meet the alignment requirements. The lateral position and rotation angle of the second metasurface are adjusted until all markers match perfectly. Through the above alignment process, the lateral displacement deviation, the rotation angle deviation and the tilt angle deviation can be controlled below $1\ \mu\text{m}$, below 0.1° and below 0.1° , respectively. After this alignment process, slight adjustment will be further performed to make sure the purity and efficiency of the OAM light are already optimal.”

Comment 2: The authors detailed in Fig. 2 that "the two metasurfaces are situated back-to-back, with an intermediate area consisting of a 1.1 mm thick silica substrate and a 0.2 mm air gap". However, it seems there was a discrepancy between the conceptual design and the actual implementation of the measurements. Could you elucidate why these differences exist? Additionally, the authors mentioned the potential for integrating the two metasurfaces to mitigate error, yet there is no initial measurement data provided on the integrated metasurfaces (e.g. LG beam profiles, purity or efficiency). Could you clarify this issue?

Our response: We thank the reviewer for this careful review and important question. The pre-version of manuscript had a spelling error and should be "face-to back". In the experiment, two metasurfaces are placed face-to-back, with an intermediate distance containing of a 1.1 mm thick silica substrate and a 0.2 mm air gap. The light is incident from the substrate side of the first metasurface and finally forms the LG mode after passing through the second metasurface. We correct the mistake in Fig. 2, Page 4. In response to the reviewer's requirements, the integrated cascaded metasurfaces with SU-8 photoresist as the spacing layer are fabricated and further characterized. The fabrication process is shown in Fig. R2. The first TiO₂ metasurface is fabricated following the standard top-down process. Then 10- μ m thick SU-8 layer is spin-coated under negative pressure to ensure fully filling the nanopillars gaps. Then the Si₃N₄ film is deposited using Plasma-Enhanced Chemical Vapor Deposition (Oxford PlasmaPro 800Plus). A new set of markers are fabricated on the Si₃N₄ film. The alignment errors between top and bottom layer markers are characterized by microscopy and compensated during the second electron-beam lithography exposure.

Figure R2 (Figure S13 in the Supplementary Materials). The fabrication process of the integrated cascaded metasurfaces.

The optical images and SEM images of the integrated metasurfaces are shown in **Fig. R3** (a) and (b). Smooth and vertical sidewall are obtained for both metasurfaces. The optical characterization results are shown in **Fig. R3** (c) and (d). For the generation of LG mode with $[l, p] = [200, 10]$, the integrated cascaded metasurfaces result in an overall purity of **86.03%** and efficiency of **70.80%**, which are higher than the results in the main text (overall purity: 82.66%, efficiency: 70.48%). The intensity profile inset

in Fig. R3(d) also shows typical rings shaped LG mode and matches very well with the simulation. And following the efficiency and purity of vortex beam versus the lateral displacement between two metasurfaces shown in Fig. S15 in the Supplementary Materials, the specific lateral misalignment is 134 nm. Further fabrication process optimization can be performed to decrease the lateral misalignment in experiment. Considering the compactness and the convenience, the integrated cascaded metasurfaces show great advantages for generating LG modes. We would improve the results in the future work and use same materials for both metasurfaces (priority using Si_3N_4).

Figure R3 (Figure S14 in the Supplementary Materials). Experimental generation of the $LG_{10,200}$ mode with integrated cascaded metasurfaces. (a), (b) The top-view optical microscope images of two metasurfaces. The insets are their high-resolution tilt-view SEM images. (c) and (d) show the experimentally recorded weighting coefficients along the Azimuthal index (l) and radial index (p), respectively. The orange dots in (c) and (d) are the numerically calculated weighting coefficients on Azimuthal and radial indices. The insets in (d) display the recorded beam profile at far field.

The corresponding discussion has been added in Para-2, Page-7 of the revised manuscript.

“Notably, a global design process that takes into account all possible experimental errors helps to address the challenge in alignment. By directly integrating two metasurfaces into a single chip, all the alignment problems can be overcome by the mature nano-alignment technology. As depicted in Supplementary Note-9, the monolithically integrated cascaded metasurface shows higher compactness and higher

conversion efficiency.”

We also added the related contents in the Note-9: of the Supplementary Materials.

“The integrated cascaded metasurfaces with SU-8 photoresist as the spacing layer are also fabricated and further characterized. The fabrication process is shown in Fig. S21. The first TiO₂ metasurface is fabricated following the standard top-down process. Then 10-μm thick SU-8 layer is spin-coated under negative pressure to ensure fully filling the nanopillars gaps. Then the Si₃N₄ film is deposited using Plasma-Enhanced Chemical Vapor Deposition (Oxford PlasmaPro 800Plus). A new set of markers are fabricated on the Si₃N₄ film. The alignment errors between top and bottom layer markers are characterized by microscopy and compensated during the second electron-beam lithography exposure. The optical images and SEM images of the integrated metasurfaces are shown in Fig. S22 (a) and (b). Smooth and vertical sidewall are obtained for both metasurfaces. The optical characterization results are shown in Fig. S22 (c) and (d). For the generation of LG mode with $[l, p] = [200, 10]$, the integrated cascaded metasurfaces result in an overall purity of 86.03% and efficiency of 70.80%, which are higher than the results in the main text (overall purity: 82.66%, efficiency: 70.48%). The intensity profile inset in Fig. S22(d) also shows typical rings shaped LG mode and matches very well with the simulation. Considering the compactness and the convenience, the integrated cascaded metasurfaces show great advantages for generating LG modes.”

Comment 3: In the authors' response to the other reviewer, they mentioned the potential benefits of utilizing higher order LG beams for the formation of orthogonal OAM states. However, taking into account the misalignment and fabrication issues discussed above, I harbor doubts concerning the actual orthogonality achieved in practice. I would appreciate further elaboration on this matter.

Our response: We thank the reviewer for this important concern. The reviewer is right. In principle, the cascaded can generate higher order LG beam with arbitrary topological charge number and orthogonal OAM states can be obtained. However, the misalignment and fabrication issues indeed will slightly degrade the purity of OAM beam, and affect the orthogonality. In our experience, the cascaded metasurfaces are integrated onto a chip and fundamentally solve the misalignment problem. As shown in Fig. R3, the purity in p, l can be achieved up to 93.20% and 92.31%, respectively. At the same time, the weight coefficients are 1.44%, 3.37%, 2.40% and 0.48% for $l = 198, 199, 201$ and 202 , while the weight coefficients are approximating to zero for other modes. As a result, only small cross-talk between neighboring OAM modes and high orthogonality can be maintained.

The intensity and the phase of the remaining Gaussian beam which is not converted to vortex generation after the second metasurface are also plot in **Fig. R4** in simulation. The region numbered as 2 is the area that vortex light is mainly concentrated on and the intensity is 88.31% with perfect helical phase. Then power in the region 1 and 3, which is inside and outside the vortex generation is 2.39% and 9.31%. The phase distributions are random and focused phase for region 1 and 3 at the same time. The intensities are low and the phase are completely different from the helical phase. So, the light in region 1 and 3 will not degrade the orthogonality.

Figure R4. Numerical simulation for the purity of the $LG_{10,200}$ mode, including the intensity profiles (top) and the phase distributions of modulated light at regions 1-3.

The imperfect fabrication will also degrade the preformation of the vortex beam. If the geometries of meta-atoms randomly deviate from the design, phase perturbation will

be induced. The random phase perturbation is introduced to both designed phase profile by adding a random matrix from 0 to 1 multiplied by a random phase amplitude δ_φ in the simulation. As shown in Fig. R5, the cascaded metasurfaces are highly robust to random scattering/phase perturbation. The efficiency only drops by 24% when the random phase perturbation even reaches 0.5π . This phase change corresponds to 25 nm dimension change, which is far larger than the experimental bias.

Figure R5. Numerical simulation for the effect of the imperfect fabrication for generation of the $LG_{10,200}$ mode. (a) shows the intensity profiles (top) and the phase distributions of modulated light. (b) and (c) show the efficiency and purity of vortex beam versus the angle variation between two metasurfaces.

Overall, our cascaded metasurfaces can conveniently obtain high-purity and high-orthogonality LG modes and would have great potential for applications like optical communications, quantum communications, and micro-manipulation.

REVIEWERS' COMMENTS

Reviewer #1 (Remarks to the Author):

Through a number of numerical simulations, sample preparations, and characterizations, the authors have thoroughly addressed each and every issue. I have no additional technical concerns, so I am pleased to recommend the publication in Nature Communications.